# The *Drosophila simulans* Genome Lacks the *crystal*-*Stellate* System

**DOI:** 10.3390/cells11233725

**Published:** 2022-11-22

**Authors:** Anna De Grassi, Patrizia Tritto, Valeria Palumbo, Maria Pia Bozzetti, Maria Francesca Berloco

**Affiliations:** 1Dipartimento di Bioscienze, Biotecnologie e Ambiente, Università Degli Studi di Bari “Aldo Moro”, 70125 Bari, Italy; 2Dipartimento di Biologia e Biotecnologie “C. Darwin”, “ Sapienza”, Università di Roma, 00185 Roma, Italy; 3Dipartimento di Scienze e Tecnologie Biologiche ed Ambientali, Università del Salento, 73100 Lecce, Italy

**Keywords:** Drosophila, heterochromatin, *cry-Ste*, evolution

## Abstract

The *cry-Ste* system is a genetic interaction system between heterochromatin and euchromatin in *Drosophila melanogaster*, regulated via the piRNA pathway. Deregulation of this system leads to meiotic defects and male sterility. Although the *cry-Ste* system is peculiar to *D. melanogaster*, ancestors of *Ste* and *Su(Ste*) elements are present in the three closely related species, *D. simulans*, *D. sechellia*, and *D. mauritiana*. The birth, evolution, and maintenance of this genetic system in *Drosophila melanogaster* are of interest. We investigate the presence of sequences homologous to *cry* and *Ste* elements in the simulans complex and describe their chromosomal distribution. The organization and expression of *cry*- and *Ste*-like sequences were further characterized in the *D. simulans* genome. Our results allow us to conclude that the *cry-Ste* genetic interaction system is absent in the *D. simulans* genome.

## 1. Introduction

The *Stellate (Ste)* gene in *D. melanogaster* maps on the X chromosome and encodes, in the male germline, for a protein called Stellate (STE), homologous to the β-subunit of casein kinase II [1,2]. The *Ste* gene expression is permanently silenced in wild-type flies, through an RNA interference mechanism [3,4]. Derepression of *Ste* gene occurs only in conjunction with specific genetic backgrounds, and when the STE protein is produced, male flies exhibit defects in meiotic chromosome condensation/segregation, show crystalline aggregates in their spermatocytes, and, in many cases, are completely sterile. All these defects were initially observed in X/0 males [5,6,7].

Genetic, cytological, and molecular dissection identified *Ste* as a component of a heterochromatin–euchromatin interaction system known as *crystal-Stellate* (*cry-Ste*). This system is a multicopy gene family, composed of two distinct *Ste* loci on the X chromosome and a third locus on the Y chromosome, initially called *crystal* (*cry*) based on the crystalline aggregates phenotype, then renamed *Suppressor of Stellate Su(Ste)* referring to the function (Figure 1). *Ste* and *Su(Ste)* are paralogous sequences organized as clusters of repeated sequences, with different degrees of repetition [8]. *Su(Ste)* keeps the normally silent status of the X-linked *Ste* through a PIWI-interacting RNAs (piRNA) pathway, a small RNA-silencing pathway involved in the transcriptional and post-transcriptional silencing of transposons and repetitive elements. In the male germline, the *cry* locus produces sense and antisense transcripts which form double-strand non-coding RNAs, precursors of piRNAs of 23 to 29 nucleotides essential for *Ste* silencing [3]. In X/0 flies or in males lacking the Y-linked *cry* locus (X/Y*^cry1^*), the absence of *Su(Ste)*-derived piRNAs (*Su(Ste)*-piRNAs), allows the synthesis of STE in the primary spermatocytes, displayed as crystalline aggregates, and leads to the mutant phenotype. A similar phenotype has also been observed when the piRNA pathway is deregulated [9,10,11]. In fact, mutations in several genes involved in the piRNA pathway regulation fail to silence *Ste*, leading to STE synthesis in spermatocytes and to the associated defects [12,13,14].

### 1.1. Structural Organization of the crystal-Stellate System

Recently, a deeper sequence analysis of *D. melanogaster* heterochromatin-enriched assemblies has produced a more accurate definition of the *cry-Ste* genomic structure, previously based only on Southern blotting and restriction map analysis. The new sequencing data broadly confirm the known X-linked *Ste* structure. *Ste* genes are organized in two clusters on the X chromosome: the euchromatic *Stellate (Ste^eu^)* locus in the 12E1,2 region of the cytogenetic polytene map as a tandem repeat of about 12 copies, and the heterochromatic *Stellate* (*Ste^het^*) locus in the h27 region of the mitotic prometaphase heterochromatin map, including about 20 copies (Figure 1) [1,5,6,8,15]. Each *Ste* unit is about 1250 bp, and genes in both clusters contain ORFs of 750 nt [7,16]. Our previous studies on *D. melanogaster* natural populations depicted a great variability of the euchromatic *Ste* locus ranging from 15–50 copies, corresponding to the *Ste^+^* allele, to about 150–400 copies corresponding to the *Ste* allele. Both the crystals’ shape and the severity of the meiotic phenotype are related to the *Ste* allelic state. In the presence of the *Ste* high copy number allele, males lacking the *cry* locus produce star-shaped crystals, their meiotic defects are more severe compared to the *Ste^+^* allele, and they are completely sterile [7,8]. Cytological analysis of male meiosis in *Ste^+^* testis revealed that meiotic chromosomes appear morphologically normal but occasionally display altered chromosomes segregation; in the *Ste* males, the chromosomes are abnormally condensed and, in many cases, appear to be fragmented [7,8].

The *Su(Ste)* locus, in the h11 region of the mitotic metaphase map of the Y chromosome, was previously reported as a cluster of about 80 units that could reach up to 240 copies in natural population strains [17,18,19]. Only recently, the detailed analysis of Y-linked sequences improved the knowledge of locus structure; in fact, 627 units of *Su(Ste)* have been identified, primarily occurring in tandem repeats and distributed in two adjacent clusters [15]. Each *Su(Ste)* unit is about 2.8 Kb and is composed of three elements: the Hoppel (1360) transposon, the region homologous to *Ste* with about 90% of nucleotide identity, and the AT-reach region specific of the element (Figure 1). Although small differences exist among the units, almost all repeats have damaged open reading frames and both strands of *Su(Ste)* units transcribe a polyadenylated mRNA in the primary spermatocytes, precursors of *Su(Ste)*-piRNAs [20].

### 1.2. Evolution of the cry-Ste System

An evolutionary history of the *cry-Ste* family has been outlined, mainly by Russian researchers, over many years. These studies suggest that the ancient precursor of *Ste* and *Su(Ste)* is the *CK2βtes* gene also known as *Ssl (Suppressor of Ste-like)*, an evolutionary conserved autosomal gene, encoding the testis-specific β-subunit of casein kinase II, mapping to the 60 D1 region of the polytene second chromosome [21]. STE and CK2βtes proteins share 53% of amino acid identity. According to this hypothesis, in *D. melanogaster*, *CK2βtes* gene has generated the *cry-Ste* system through a series of events (Figure 2). Particularly, a duplication of *CK2βtes* gene on male chromosomes generated the Y-linked pseudo *CK2βtes* repeat, which in *D. melanogaster* is named *PCKR* (Pseudo-*CK2βtes* Repeat); recently, a cluster of *PCKR* pseudogenes has been reported as a clade of 122 units, mapping close to the *cry* locus [15,22,23]. In a successive step, a copy of the Y-linked *CK2βtes* acquired a promoter region through non-homologous recombination from an X-linked gene family called *NACβtes*, driving testis-specific expression and generating the chimeric intermediate *NACβtes*-*CK2βtes* [23]. A duplication event translocated the latter ancestral copy from the Y to the X chromosome and, subsequently, the two elements underwent amplification independently in the sex chromosomes. A recent phylogenetic sequence analysis and promoter region characterization of *Ste^eu^* and *Ste^het^*, suggest that the appearance of euchromatic and heterochromatic *Ste* clusters on the X chromosome occurred and evolved independently of each other [15,24].

In the Y-linked chimeric element, the insertion of the Hoppel transposon and subsequent amplifications generate *Su(Ste)* [25]. Hoppel provides a promoter and allows *Su(Ste)* to produce the double-stranded RNAs that generate piRNAs responsible for *Ste* suppression [3].

From a chronological point of view, the described evolutionary events occurred in a common ancestor of *D. melanogaster*, *D. simulans*, *D. sechellia*, and *D. mauritiana*, but the *cry-Ste* system emerged in *D. melanogaster* only [25].

A *cry-Ste* system, such as that in *D. melanogaster*, seems to be absent in closely related species, although sequences homologous to both elements, here called *Ste*-like and *Su(Ste)*-like, reside in their genomes. Livak [8] reported the presence of *Ste*-like sequences in *D. melanogaster* sibling species, *D. simulans* and *D. mauritiana*, with a reduced copy number and localized only on the *Y* chromosome. In a de novo assembly of the simulans species complex, copies of *CK2βtes-Y* corresponding to *PCKR* have been identified on the Y chromosomes of *D. simulans*, *D. mauritiana*, and *D. sechellia* (92, 117, and 22 copies, respectively) [15,26,27]. The absence of the *cry-Ste* system and the analysis of interspecies hybrid sterility has been recently reported for *D. mauritiana* [28,29].

The evolutionary and functional significance of the *cry-Ste* system is still an enigma, and the reasons for its unique selection in *D. melanogaster* remain to be elucidated.

*D. melanogaster* has three closely related species, *D. simulans*, *D. sechellia*, and *D. mauritiana.* All four species are very similar morphologically and are collectively known as the Drosophila melanogaster complex [30]. These species are reproductively isolated from each other [31,32,33]. In this study, we investigate the presence of the *cry-Ste* elements in *D. simulans*, *D. sechellia*, and *D. mauritiana*. The latter three species constitute the Drosophila simulans complex. In this study, we have isolated species-specific *Ste*- and *Su(ste)*-like sequences and analyzed their chromosomal distribution. We have also provided a detailed view of the organization of these sequences in *D. simulans* genome. Our expression analysis shows that in *D. simulans* a transcript compatible with the 750 nt Ste mRNA is not present. Collectively, we can conclude that the *cry-Ste* system is absent in the *D. simulans* genome.

## 2. Materials and Methods

### 2.1. Fly Strains and Genetic Crosses

Flies were grown on standard cornmeal-agar medium at 25 °C.

*D. melanogaster (Oregon-R)*, *D. simulans* (S130), *D. sechellia* (S-9), and *D. mauritiana* (S080), strains present in the collection at our department, were obtained from Umea Drosophila Stock Center (Sweden). *D. simulans C(1;Y)A13/C(1)RM,y w/0* was kindly provided by John Roote.

Genetic crosses: to obtain *X/0* males, *D. melanogaster* and *D. simulans*, X/Y males were crossed to *C(1)RM*, *y*^2^
*su(w^a^) w^a^/0*, and *C(1)RM,y w/0* females, respectively, and F1 males were collected.

### 2.2. Fluorescence In Situ Hybridization (FISH)

Polytene chromosomes were prepared according to Pardue [34]. Chromosome metaphase preparations and FISH experiments were carried out as described previously [35,36]. Probes were directly labeled with Cy3-dUTP, Cy2-dUTP, or FluorX-dCTP (Amersham). Preparations were counterstained with 4′,6-diamidino-2-phenylindole (DAPI). Digital images were recorded using an Olympus epifluorescence microscope equipped with a cooled CCD camera. Fluorescent hybridization signals were pseudocolored and merged using the Adobe Photoshop software.

### 2.3. Molecular Analysis

DNA fragments from restriction digest of phage DNA and from polymerase chain reaction (PCR) were cloned into the pGEM7 Vector and pGEM-T Vector (Promega), respectively, and analyzed by DNA sequencing. The sequences of the obtained clones are listed in Appendix A.

Primers used for cloning: Up1Su(Ste) 5′-TAGGGATGACGTGATTTTTTAA-3′; Lw2Su(Ste) 5′-TCCGGTCAGGGTTTCCTTCGTAA-3′; Up3Su(Ste) 5′-TTTCCAAAAGTGTTTTCCTGA-3′; Lw3 Su(Ste) 5′-AAATGCACTGCCAGTTTT-3′; G25 Ste 5′-GTGCAACCGAAAGCCTAGGTA-3′; G26 Ste 5′-GTGAACTGGCAACATGTCGAG-3′.

*Su(Ste)*-specific 663 clone was generated by PCR amplification of the genomic DNA cloned in the λEMBL3 vector, named Dmλ16.1, using the primers Up1Su(Ste)-Lw2Su(Ste). Dmλ16.1 phage belongs to a collection of lambda phages, molecularly characterized previously, selected by a screening of *D. melanogaster* genomic library, using pUSTE6 as a probe [37]. The cloned DNA of 663 nt corresponds to the Z11734 element of the *cry* repeat (nucleotides 2146-2791) [10].

Clones called cryme3 (386 nt), crysi3 (380 nt), cryse3 (382 nt), and crymau3 (389 nt) come from the PCR amplification (using the primers Up3-Lw3) of genomic DNA from *D. melanogaster*, *D. simulans*, *D. sechellia*, and *D. mauritiana* males, respectively (Appendix A).

The genomic libraries (in λEMBL3 vector) of *D. melanogaster* and *D. simulans* are described in [38]. Positive plaques from genomic library screening were isolated using standard screening protocol [39]. pUSTE6 and crysi3 were used as probes to screen the *D. simulans* genomic library.

Clones were obtained by subcloning the indicated DNA restriction fragments, derived from the selected phages: the F6.3 clone, EcoRI-XbaI DNA restriction fragment of 471 bp, from the phage named DsλF6; the F4C2 clone, EcoRI-BamHI of 1041 bp, from the phage named DsλF4; the 20F1 clone, KpnI-BamHI of 775 bp, from the phage named DsλF20; the 17F7clone, KpnI-BamHI of 920 bp from the phage named DsλF7; the F39A clone, EcoRI-XhoI of 3559 bp, from the phage named DsλF39.

Southern Blotting analysis. DNA probes labeling and Southern blotting analysis were performed as described by Sambrook [39]. Membranes were stripped in boiling 0.1% SDS solution and incubated for 30 min. Prior to re-use, filters were checked for residual radioactivity.

Northern Blotting analysis. Total RNA samples were purified from about 100 testes of adult flies, with Trizol Reagent, according to the protocol supplied (Invitrogen). Northern blotting hybridization was carried out using (α^32^P)-dUTP labeled *Ste* cDNA antisense riboprobe [39]. Riboprobe was produced by T7 polymerase in vitro transcription of *Ste* cDNA, cloned in pGEM7, after digestion with Eco RI, using commercial Riboprobe in vitro transcription kit, according to manufacturer’s protocol (Invitrogen).

piRNA detection on Northern blotting was performed as described in [40]. DNA oligonucleotides complementary to the sequence of piRNAs named crystal, and 5S RNA were used as probes. Sequence of the crystal oligonucleotide: 5′-UCGGGCUUGUUCUACGACGAUGAGA-3′. Probes were 5′-^32^P-radiolabelled with polynucleotide kinase (Roche).

### 2.4. Sequences Analysis

Sequences analysis was performed by BLAST (Basic Alignment Search tool) using species data available at FB2022_03 (release 9 June 2022) from http://www.flybase.org (accessed on 1 July 2022) and/or https://blast.ncbi.nlm.nih.gov/Blast.cgi (accessed on 1 July 2022) (release 17 March 2022). Transcriptome data of *D. simulans* testes consulted by BLAST: (SRX5802613, *D. simulans* testes; NCBI Bio project PRJNA541548). The sequence similarity analysis between the F6.3 sequence and the *D. melanogaster* genome was conducted using the BLAT algorithm on the UCSC Genome Browser (https://genome.ucsc.edu/cgi-bin/hgBlat, accessed on 28 October 2022). The resulting sequences, together with the full sequences of *Ste*, *Ste^het^*, *SuSte*, and *PCKR*, weremultialigned using MultAlign (http://multalin.toulouse.inra.fr/multalin/, accessed on 28 October 2022). Alignment editing and pair-wise sequence-identity calculation were performed in Jalview (https://www.jalview.org/, accessed on 28 October 2022).

## 3. Results

### 3.1. Overview of Ste- and Su(Ste)-like Sequences in the Melanogaster Complex

To investigate the presence and organization of *Ste*- and *Su(Ste)*-like repeated elements in the melanogaster complex species, being these sequences repeated, we analyzed *D. simulans*, *D. sechellia*, and *D. mauritiana* genomes using fluorescence in situ hybridization (FISH), Southern blotting, and DNA sequencing. We have used distinct sequences arising from both *Ste* and *Su(Ste)* as a probe for both FISH and Southern blotting experiments. As *Ste* probe, we utilized the *Ste* coding sequence contained in the pUSTE6 plasmid [2], while to highlight *Su(Ste)* only, we cloned a DNA fragment of 663 nt (non-homologous to *Ste*) from the AT-rich region of *Su(Ste)*. The latter, which we called *Su(Ste)*-specific 663 (Figure 1 and Appendix A), corresponds to the z11734 element of the *cry* repeat (nt 1842-2487). It was obtained by PCR amplification of a phage named Dmλ16.1, carrying a DNA insert arising from the *D. melanogaster* Y chromosome. This phage belongs to a collection of previously molecularly characterized lambda phages, isolated by a screening of the *D. melanogaster* genomic library [37].

Both sequences were mapped by FISH on the *D. melanogaster* polytene and mitotic chromosomes. The pUSTE6 probe highlights the canonical *Ste* loci on the X and Y mitotic chromosomes (Figure 3B), chromocenter, and the *Ste^eu^* cluster in the 12E1,2 region on the polytene chromosomes (Figure 3F) [7]. The *Su(Ste)*-specific 663 probe gives a unique wide signal peculiar to the repeat, in the h11 region on the Y mitotic chromosome (Figure 3C), which, as expected, is co-localized with the pUSTE6 signal in the *Su(Ste)* locus (Figure 3D). The same probe also hybridizes, as a thin band, in the region between 12D and 12E1,2 bands of the cytogenetic map of the X polytene chromosome (Figure 3G,H).

Cytological mapping of *Su(Ste)*-specific 663 on the polytene X chromosome is consistent with genome sequence alignment data, which will be discussed below. It showed the presence of a sequence of about 350 nt, homologous to *Su(Ste)*-specific 663, located upstream of the *Ste^eu^* cluster at exactly 426 bp at 3′ side of the distal marginal *Ste* gene CG33247, in the 12D3 band of the cytogenetic map.

Note that the widespread signal due to the *Ste* repeated units spans in the adjacent band 12D, resulting in an overlap of the signals from the two probes (Figure 3).

The repetitive nature and distribution of both sequences in the genome of males and females are also detectable by Southern blotting analysis, using the same sequences utilized for the in situ experiments as probes. The hybridization of the pUSTE6 probe on the *D. melanogaster* genome confirms the previously described CfoI restriction pattern, with the three main fragments (1150, 1100, 950 bp) X-linked and the 800 bp Y-linked fragment [7] (Figure 4A). The *Su(Ste)*-specific 663 probe reveals in *D. melanogaster* males a 500 bp CfoI main fragment and additional bands with a higher molecular weight, while very few bands occur in females (Figure 4B). The latter observed pattern resembles the chromosomal distribution visualized by FISH in Figure 3, as a repeat on the Y mitotic chromosome and as a single band on the polytene X chromosome.

The presence of the two elements of the *cry-Ste* system was analyzed by Southern blotting in *D. simulans*, *D. sechellia*, and *D. mauritiana* genomes, using heterologous probes. The hybridization pattern of pUSTE6 (Figure 4A) indicates the presence of *Ste*-like sequences as moderately repetitive, and almost exclusively clustered in the Y chromosome in the three species; moreover, few bands are scattered in females. *D. simulans* and *D. sechellia* have a restriction pattern that is quite similar, while additional bands occur in *D. mauritiana*. *Su(Ste)*-specific 663 probe has also shown a different hybridization pattern in the simulans complex species compared to *D. melanogaster*: few bands with a shared restriction pattern between males and females (Figure 4B). The patterns resulting from the Southern blotting analysis, indicate that the *Ste*-like and the *Su(Ste)*-specific 663-like sequences have a different repetition degree and a different chromosomal distribution in the simulans complex species, compared to *D. melanogaster*. This suggests that the genome of the simulans complex species do not host a repeated element, comprising both sequences, similar to *Su(Ste)* of *D. melanogaster*.

### 3.2. Ste- and Su(Ste)-like Elements Chromosomal Distribution in the Melanogaster Complex

In order to investigate the *cry*- and *Ste*-like sequence organization and chromosome distribution in the sibling species, we planned to clone species-specific *Ste*-like and *Su(Ste)*-like sequences. Using suitable combinations of PCR primers (Figure 1C), we amplified and cloned a DNA sequence corresponding to the portion of the AT-rich region of *Su(Ste)*-like sequences non-homologous to *Ste*, in all the simulans complex species. As a control, we repeated the same experiment in *D. melanogaster*, obtaining many clones that overlap with the *Su(Ste)*-specific 663 clone. However, some nucleotides were different due to the fact that small differences exist in the nucleotide sequences among the repeats. Among them, we chose a clone called cryme3 for further experiments.

Clones called crysi3, cryse3, and crymau3 come from the PCR amplification (using the primers Up3-Lw3) of genomic DNA from *D. simulans*, *D. sechellia*, and *D. mauritiana* males, respectively (Appendix A).

No amplification product, under several experimental conditions, was obtained for *Ste*-like sequences in the simulans complex species, suggesting that the sequences of the chosen primers are not present in these species.

Moreover, we have used our clones for cytogenetic analysis. In situ hybridization of cryme3, crysi3, cryse3, and crymau3 on the polytene chromosomes of the respective species provided a single hybridization signal in all species on the X chromosomes band, syntenic with the 12D region of *D. melanogaster* polytene chromosome. Examples of FISH on the polytene chromosomes of *D. simulans* and *D. mauritiana* are shown in Figure 5.

FISH experiments of the same species-specific probes on the respective mitotic chromosomes gave a hybridization signal in *D. melanogaster* while no signal was observed in the other species. Moreover, in *D. melanogaster* the cohybridization of cryme3 and *Su(Ste)*-specific 663 probes indicates their co-localization in the h11 region on Y chromosome (Appendix A). The lack of crysi3, cryse3, and crymau3 hybridization on the mitotic chromosomes of simulans complex species suggests a non-repetitive nature for these sequences, whose abundance is below the level detectable by FISH.

Cytological mapping of crysi3 and cryse3 on the polytene X chromosome of *D. simulans* and *D. sechellia* was in accordance with sequence alignment data with the respective sequenced genome and showed the presence of the cloned sequence in contigs belonging to the X chromosome (GB: CM2002914; GB: CH480835).

FISH results and Southern blotting analysis suggested that *Ste*-like and *Su(Ste)*-like sequences in the simulans complex had a different chromosomal distribution compared to *D. melanogaster*. The sequence that appears repeated in the simulans complex was identified by the *Ste* probe and the repetitions were mainly located on the Y chromosome. The *Su(Ste)*-like specific region, isolated as crysi3, cryse3, and crymau3 clones, had the same distribution in both sexes of all simulans complex species and was not repeated as in *D. melanogaster*. This last result is in agreement with the Southern blotting data in Figure 4, showing the distribution of *Su(Ste)*-specific 663-like in the simulans complex species.

### 3.3. Ste- and Su(Ste)-like Sequences in D. simulans

#### 3.3.1. Ste-like Sequences

We deepened the analysis of *(Ste)*-like and *Su(Ste)*-like sequences in the *D. simulans* genome. In order to isolate whole *(Ste)*-like and/or *Su(Ste)*-like elements, not having obtained *Ste*-like clones by PCR amplification, we carried out a screening of a *D. simulans* genomic library. pUSTE6 and crysi3 were used as probes to screen the *D. simulans* genomic library.

Using pUSTE6, we isolated about one hundred clones in lambda phages. By Southern blotting analysis of the isolated DNA, digested with CfoI restriction enzyme, most of the clones presented a single hybridization band of about 100 nt, as shown in the example in Figure 6A, while only three phages generated hybridization bands of different molecular weights. Among the subclones produced and sequenced, clones containing sequences homologous to *Ste* were molecularly characterized.

Among the phages with the same hybridization pattern, a phage named DsλF6 was selected (Figure 6A, lane 2); from this phage, we obtained a clone of 473 nucleotides, named F6.3, used as a probe in the Southern blotting and FISH analyses.

Using a Blast search, the F6.3 sequence identifies identical matches in the *D. simulans* reference genome. A maximum likelihood phylogenetic tree shows that the F6.3 sequence clusters with all the Y-linked *CK2βtes*-Y genes of the *D. simulans* reported in [27]. Moreover, the F6.3 sequence identifies identical matches with sequences corresponding to *CK2βtes-Y* transcripts in the transcriptome data of *D. simulans* testes (SRX5802613). Furthermore, the F6.3 sequence presents homology (about 75–80%) with the *Ssl* gene on 2R and with a region in the 12D band on the X chromosome (GB: CM002914) in the *D. simulans* genome. In addition, multiple hits on Uchromosomes were obtained.

According to a Blast search, no identical matches exist between the F6.3 sequence and the *D. melanogaster* genome; instead, multiple hits (with homology of 70–80%) are on Y and X chromosomes scaffolds corresponding to *Ste*, *cry*, and *PCKR* repeats (Appendix A).

The hybridization of the F6.3 clone on the filter employed in the previously Southern blotting showed a strong, wide signal on a single band of about 100 nt clearly evident in *D. simulans*, *D. sechellia*, and *D. mauritiana* males, with additional bands in particular in *D. mauritiana* (Figure 6B). The presence in the F6.3 sequence of multiple cutting sites for the restriction enzyme CfoI generated on the Southern blot a hybridization band of about 100 nt. Hybridization is also present in *D. melanogaster*, mainly on the Y chromosome. The resulting pattern suggests that in all species, the isolated DNA corresponds to DNA present in several copies on the Y chromosome.

The clone F6.3 has been mapped by FISH on the *D. simulans* polytene and mitotic chromosomes. On the polytene chromosomes, it hybridizes in the chromocenter and on 2R in the 49AB region; under low stringency conditions, hybridization signals also occur in 12D and 60D (*Ssl* gene) regions (data not shown). On the mitotic chromosome, referring to the map of *D. simulans* heterochromatin [41], we mapped a strong hybridization signal in the h15 band of the short arm of the Y chromosome and a second faint signal on the long arm, in the h11 band proximal to the centromere; the latter is visible only with a longer exposure time (Figure 7). In the in situ co-hybridization on *D. simulans* mitotic chromosomes, the F6.3 and pUSTE6 probes hybridized in the same locus on the Y chromosome (Appendix A).

Southern blotting and FISH analyses indicated a repetitive nature in the F6.3 clone sequence and a chromosome distribution peculiar to the simulans complex species. In *D. simulans*, a major cluster of this repetitive sequence was localized on the short arm of the Y chromosome and a second cluster, with few copies, mapped near the centromere. Sequence alignment of the F6.3 sequence on *D. simulans* reference genome indicated its correspondence with the *CK2βtes*-Y gene.

Molecular characterization of the three lambda phages with a different restriction pattern compared to phage DsλF6, as shown in Figure 5, enabled us to isolate three new clones (Appendix A). Analysis of the cloned sequences by Blast search with *D. simulans* and *D. melanogaster* genome references, indicate that all phages contain the *CK2βtes-Y* gene. In detail, the subclone of phages DsλF4, named F4C2, contains the *CK2βtes-Y* gene. In the phage DNA insert, the identified sequences were arranged as follows: the F4C2 sequence was adjacent, in subsequent order to an incomplete telomere-associated element TART-A, a Hoppel/1360 transposon, and the other telomere-associated element Het-A. A subclone of phage DsλF20 named 20F1 contains an incomplete *CK2βtes-Y* gene, which in the phage insert was adjacent, in subsequent order, to an incomplete Hoppel/1360 element and a Het-A transposon. The subclone of phage DsλF7, named 17F7, contained an incomplete *CK2βtes-Y* that in the phage insert was adjacent to a Hoppel/1360 transposon. Molecular analysis indicated that the DNA inserts of the three phages come from non-overlapping genomic regions. The presence in the isolated phages of the telomeric elements Het-A and TART-A, which in our previous mapping we have located in the same heterochromatic region [42], could suggest the origin of the phages insert DNA from the Y chromosome.

#### 3.3.2. *Su(Ste)*-like Sequences

All cloned sequences isolated using the *Ste* probe in the screening of *D. simulans* genomic library correspond to the *CK2βtes-Y* gene. None of the already isolated phages contain an entire element similar to *Su(Ste)*. To determine if *D. simulans* genome harbors a complete *Su(Ste)*-like element, we repeated the screening of the same genomic library, using the species-specific clone crysi3 as a probe. Unlike the screening by pUSTE6 probe, in this one we obtained two phages only. By molecular characterization, results from both phages were identical; they hybridize, although with less intensity, to pUSTE6, and probably for this reason we have missed them in the previous screening. By sequence analysis of DsλF39 phage subclones, we selected the clone named F39A, containing a DNA restriction fragment of about 3.5 Kb (Appendix A). The clone F39A includes the crysi3 sequence and in a Blast search on the *D. simulans* reference genome, it corresponded to the same region mentioned above containing the crysi3 sequence, syntenic with the 12D band of the X chromosome polytene map of *D. melanogaster*. As shown in Figure 8, it includes part of the *Su(Ste)*-specific-like region and 771 bp downstream, an incomplete *CK2βtes*-like sequence, previously defined as a *ΨCK2βtes* by Kogan et al. [25]. In a Blast alignment on the *D. melanogaster* reference genome, the best hit of F39A was as follows: it started upstream of *Ste* cluster, exactly 446 nt at the 3′ side of Ste:CG33247, and ended downstream of the *Ste* cluster, 161 nt at the 5′ side of Ste:CG33236 (Figure 8). A comparison of the syntenic X-chromosome regions of *D. simulans* and *D. melanogaster* is schematically represented in Figure 8.

The F39 clone is the only one, among all the isolated genomic regions of *D. simulans*, to contain the portion of *Su(Ste)*-specific-like region (orthologous of the AT-rich region of *Su(Ste)*) and the *CK2βtes*-like, very close to each other. Although the two sequences *Ste*-like and *Su(Ste)*-like are close together, they do not have the canonical structure of the *Su(Ste)* element. The cloned region corresponds to the site of the X chromosome where the locus *Ste* in *D. melanogaster* is located.

### 3.4. Expression of Ste-like Sequences in D. simulans

*Ste* genes produce in the testes of X/0 or X/Y*^cry1^* males of *D. melanogaster*, a polyadenylated transcript of 750 nt that is translated in the STE protein forming crystals [1,10].

We checked for the presence of a *Ste*-like mRNA in *D. simulans*. Total mRNA extracted from females, males XY, and X/0, was subjected to Northern blotting analysis, using *Ste* cDNA antisense as probe. As shown in Figure 9, the *Ste* transcript of 750 nt was present only in *D. melanogaster* X/0 males and it was absent in *D. simulans*. The ubiquitous larger 1400 and 1300 nt transcripts normally present in both females and males, and in somatic tissues, independent of *cry* locus [9,43], were found in all samples.

In the X/Y testes of *D. melanogaster* males, piRNAs that normally silence *Ste* genes are produced. To check if piRNAs corresponding to *Su(Ste)*-like sequences are present in *D. simulans*, we analyze the expression of one of the *cry*-specific piRNA, named crystal in *D. melanogaster* [40]. RNA from testes of *D. simulans* XY males were analyzed by Northern blotting using the oligonucleotides Su(Ste)4 as a probe [40]. No hybridization signals were detected, except in XY males of *D. melanogaster* used as a control (Appendix A).

The Northern blotting analysis clearly indicates that in *D. simulans* testes, neither a *Ste*-like transcript of 750 nt nor crystal-like piRNAs are produced.

We also analyzed testes in X/0 males of *D. simulans* by immunostaining, using a TS1 anti-STE antibody [2]. In the testes no crystalline aggregates were detected by immunostaining. The absence of STE protein in the examined tissue confirms that the X-linked *Ste*-like sequences do not transcribe for a coding *Ste*-like transcript of 750 nt in *D*. *simulans* testes.

## 4. Discussion

### 4.1. cry-Ste Origin in D. mel-sim Clade

The evolutionary significance of the *cry-Ste* system and the reasons for its unique selection in *D. melanogaster* are still open questions. Our findings clarify some aspects and are in agreement with previous observations about the presence of sequences homologous to *Ste*- and *Su(Ste)*- genes in *D. simulans*, *D. sechellia*, and *D. mauritiana* [8,27,29,44,45]. All *Ste*-like clones isolated from the *D. simulans* genome contain repeated sequences corresponding to the *CK2βtes-Y* gene, or *PCKR* in *D. melanogaster*. A major cluster of this repetitive sequence is localized on the short arm, in the h15 band, of the Y chromosome and a second cluster, with few copies, near the centromere. Our data explain that the pattern observed by the Southern blot analysis of *Ste*, shown here and previously reported [8], is due to the hybridization of *Ste* to the *CK2βtes-Y* gene. Southern blotting analysis indicates the presence of its homologous sequences on the Y chromosome in *D. sechellia* and *D. mauritiana* and with a similar abundance. Southern blotting and FISH data indicate that no *Su(Ste)*-like repeat occurs on the Y chromosome of simulans complex species. All analyzed sequences from *D. simulans* library, carry the *CK2βtes-Y* gene adjacent to transposable elements, such as Hoppel, Het-A, or TART, but none of them exhibit a *Su(Ste)*-like structure. The one clone harboring an incomplete *Ste*-like sequence, more precisely *ΨCK2βtes-Y*, near a *Su(Ste)* specific-like sequence isolated in *D. simulans*, comes from the region of the X chromosome, corresponding to the site where a *Ste* euchromatic cluster arose in *D. melanogaster* (Figure 8). Expression analysis shows that in *D. simulans* no *Ste*-like transcript of 750 nt is produced in the testes of X/0 males, and also no crystalline aggregates are detected in their testes by immunostaining using the TS1 anti-STE antibody. Furthermore, Northern blotting analysis revealed the absence in *D. simulans* males, of one of the abundant piRNAs of *D. melanogaster* testes, called crystal. The results of our search for *Ste*-like and *Su(Ste)*-like sequences suggest that in *D. simulans* there are neither *Ste* genes nor *Su(Ste)* repeats; therefore, we can conclude that the genome of this species lacks the *cry-Ste* genetic interaction system.

Our data, demonstrating the absence of the *cry-Ste* system in *D. simulans*, in addition to data in *D. mauritiana* reported by Adashev et al. [29], support the fact that the *cry-Ste* system is fixed only in the *D. melanogaster* genome [8,24,25,26,27]. Chromosomal location of the isolated *Ste*-like sequences, on the Y chromosome of the analyzed species, argues the current hypothesis that a duplication and subsequent amplification on the Y chromosome of *CK2βtes-Y* occurred in a common ancestor of the melanogaster complex species [15,24,25]. The subsequent steps, which are hypothesized to have occurred only in *D. melanogaster* after the divergence of simulans complex, led to the amplification of *Ste* on the X chromosome and the birth of *Su(Ste)* on the Y chromosome [25].

The same organization of the sequences observed in the F39A clone of *D. simulans*, in the region correspondent to *Ste^eu^* cluster in *D. melanogaster* (Figure 8), is found also in the orthologue region in *D. sechellia* (data not shown); these findings support the hypothesis that *Ste* amplification occurred in *D. melanogaster* after divergence of the simulans complex. Whether these sequences were the background for the birth of the *Ste^eu^* repeat, and what events generate the amplification of *Ste* only in *D. melanogaster*, has not yet been clarified. As already mentioned, in natural populations of *D. melanogaster*, there exists a great variability in the *Ste* copies number in this locus. Although the frequency of the *Ste^+^* allele (89%) exceeds that of the *Ste* allele, flies with up to 400 copies of *Ste^eu^* have been isolated [7]. Analysis of the chromosomal distribution of *Ste*-like ancestor sequences could be useful to reconstruct the evolutionary history of the *cry-Ste* system. The portion of *Su(Ste)*-specific-like region of about 400 bp identified in the F39A (Figure 8), orthologous to the AT-rich region of *Su(Ste)*, is present in the same chromosomal location on the X chromosome in both *D. sechellia* and *D. mauritiana*. This common chromosomal situation could be suggestive of a role of this region as an intermediate site for *Su(Ste)*’s birth before its amplification on the Y chromosome in *D. melanogaster.*

### 4.2. The Biological Function of the cry-Ste System

In *D. melanogaster*, the *cry-Ste* system is normally silenced in XY males, so we can argue that this system is dispensable for male fertility. Citing Palumbo et al. [7], “*cry-Ste* seems to be a selfish genetic system whose components are normally silent, but if perturbed cause perturbation of normal meiotic processes in which is intimately nested”. The *Su(Ste)* sequences work to suppress expression of their homologous sequences. The reasons that favored the creation and, above all, the maintenance of this genetic X–Y interaction system have not yet been understood.

As described, loss of *Su(Ste)* causes elevated levels of nondisjunction, low fertility, or sterility depending on the *Ste* allele and also production of distorted sperm genotype ratios (meiotic drive) towards female offspring [7]. There are different points of view in the consideration of the *cry-Ste* system as a case of meiotic drive. A hypothesis has been proposed that this is a cryptic drive system where *Ste* is a meiotic drive gene [46]. Moreover, other authors, to test the idea that *Ste* overexpression or the lack of *cry* causes the meiotic drive in the Y*^cry^*^1^ males, have demonstrated that the almost complete deletion of X-linked *Ste* copies does not eliminate meiotic drive and nondisjunction [43]. 

Recent studies of interspecies hybrid have analyzed the behavior of the *Ste* gene expression in the interspecies hybrids between *D. melanogaster* and *D. mauritiana* [28,29]. The hybrid X*^D.mel^*/Y*^D.mau^* males are sterile, and derepression of *Ste* genes and STE protein aggregates have been observed in their testes. The lack of Y-derived *Su(Ste)* piRNAs in the hybrid males allows translation of the *Ste* transcript coming from the X chromosome of *D. melanogaster.* Adashev et al. propose the *cry-Ste* as one of the causative factors of hybrid sterility and suggest a functional role for this system, through a piRNA pathway, in the reproductive isolation of a species [28,29]. Based on the lack of *cry* locus on the Y chromosome of *D. simulans*, we can assume that interspecific hybrid of *D. melanogaster* with *D. simulans* could behave like *D. melanogaster*/ *D. mauritiana* hybrid males. However, we could not carry out any cytological analysis of testes isolated from hybrid males resulting from *D. melanogaster/D. simulans* crosses because of their severely reduced size, associated with abnormal morphology and indistinguishable cell populations. Collectively, in this cytological analysis, we have encountered the same difficulties met by Adashev et al. [28,29].

The silencing of *Ste* genes, which guarantees male fertility, occurs through a piRNA pathway. In *D. melanogaster* testes, piRNAs are produced in a stage-specific manner during spermatogenesis. In spermatogonia, the vast majority are TE-mapping piRNAs, while in early spermatocytes most of the identified piRNAs arise from the *Su(Ste)* clusters. This last stage of spermatogenesis corresponds to a temporal window, with a decreased level of several components of the piRNA pathway [47]. Chen et al. recently identified *petrel*, a new Y-linked cluster, as a source of piRNAs, able to silence the homologous euchromatic gene *pirate* in the testes [48]. In mice, piRNAs mediating silencing of TEs is essential for male fertility. Recently, a source of testis-specific piRNAs have been discovered on the Y chromosome, whose deletion disrupt sperm maturation [49,50]. Understanding why the *cry-Ste* system is regulated through a piRNA-mediated pathway would be useful to understand the causes and forces driving the evolution of this system. STE protein synthesis takes over in males when two conditions occur: (a) when spermatocytes do not carry the chromosome Y or carry a Y*^cry1^* chromosome and no *Su(Ste)* piRNAs are produced; (b) when the piRNA pathway is deregulated. STE protein triggers the processes that determine the decreased fertility or the complete sterility, thus disfavoring Y chromosome-free sperms and/or sperm with a high TE mobilization. In essence, silencing of *Ste* genes ensures fertility—and, therefore, the offspring for that individual—when the conditions of genomic integrity and stability are met. If these conditions do not occur, then the Stellate phenotypes occur.

The existence of other similar genetic systems in *D. melanogaster*, in melanogaster complex, or in other species is under investigation to discover whether they are unique to *D. melanogaster* or other species have evolved the same type of regulatory systems, however, species- and sex-specific.

In the last 30 years, new mechanisms that regulate the physiology of heterochromatin have been discovered, but an idea from pioneers of this analysis can still be considered current:

“*A central issue about constitutive heterochromatin concerns the mechanisms underlying its evolutionary conservation. The presence of large amounts of heterochromatin in the germline of most higher eukaryotes, and its fine quantitative regulation in certain somatic tissues, suggest that heterochromatin is an integral part of the genome that positively contributes to fitness*”(Gatti and Pimpinelli. Pg 267, Annu. Rev. Genet. 1992).

## Figures and Tables

**Figure 1 cells-11-03725-f001:**
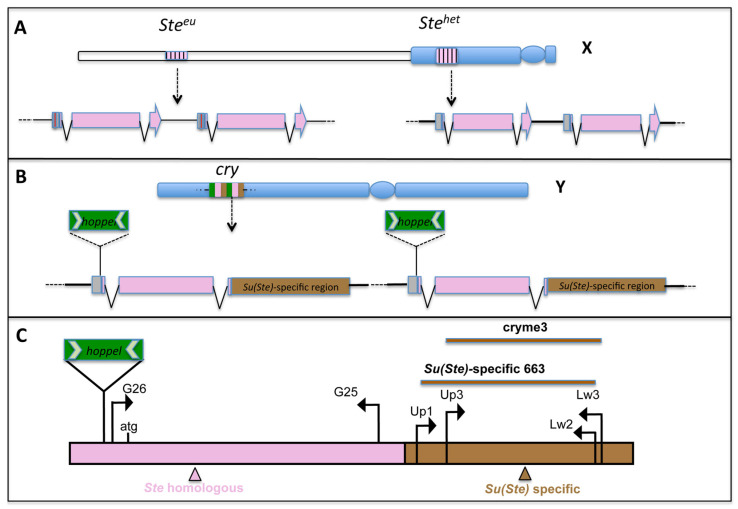
(**A**,**B**) Schematic representation of the *Ste* and *cry* loci on *D. melanogaster* sexual chromosomes. Heterochromatin is depicted in blue, euchromatin in white. The paralogous sequences of the repeats are depicted by the same color (pink); the green arrow points the insertion site of the Hoppel transposon in the *Su(Ste)* element; the *Su(Ste)*-specific region is in brown. (**C**) Position of the primers used to amplify the region-specific probe. All drawings are not to scale.

**Figure 2 cells-11-03725-f002:**
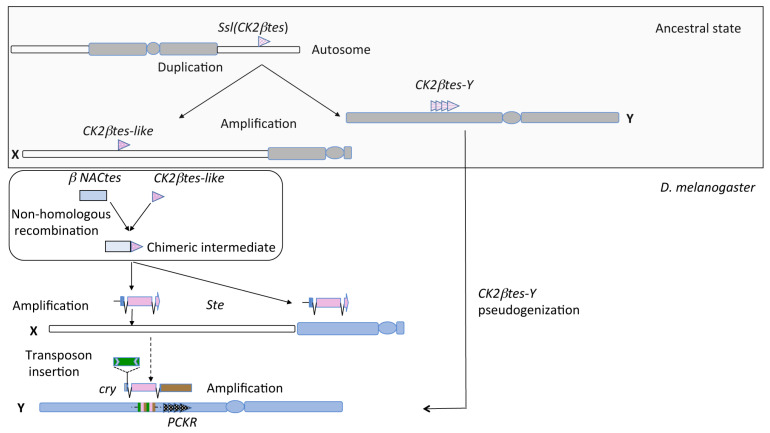
Diagram showing the inferred evolutionary history of the *cry-Ste* system. In the melanogaster complex ancestor, the *Ssl* gene translocated on the sex chromosomes. The copy translocated on the Y chromosome, amplified generating the *CK2βtes-Y* repeats. In *D. melanogaster*, on the Y chromosome a pseudogenization of *CK2βtes-Y* generated *PCKR.* A non-homologous recombination of *CK2βtes-like* gene and an X-linked gene family called *NACβtes*, generated a chimeric intermediate; a duplication event then generated the *Ste* genes that evolved independently to the *Ste^eu^* and *Ste^het^* loci on the X chromosome. The insertion of a Hoppel transposon and the following amplification on the Y chromosome finally produced *cry* genes.

**Figure 3 cells-11-03725-f003:**
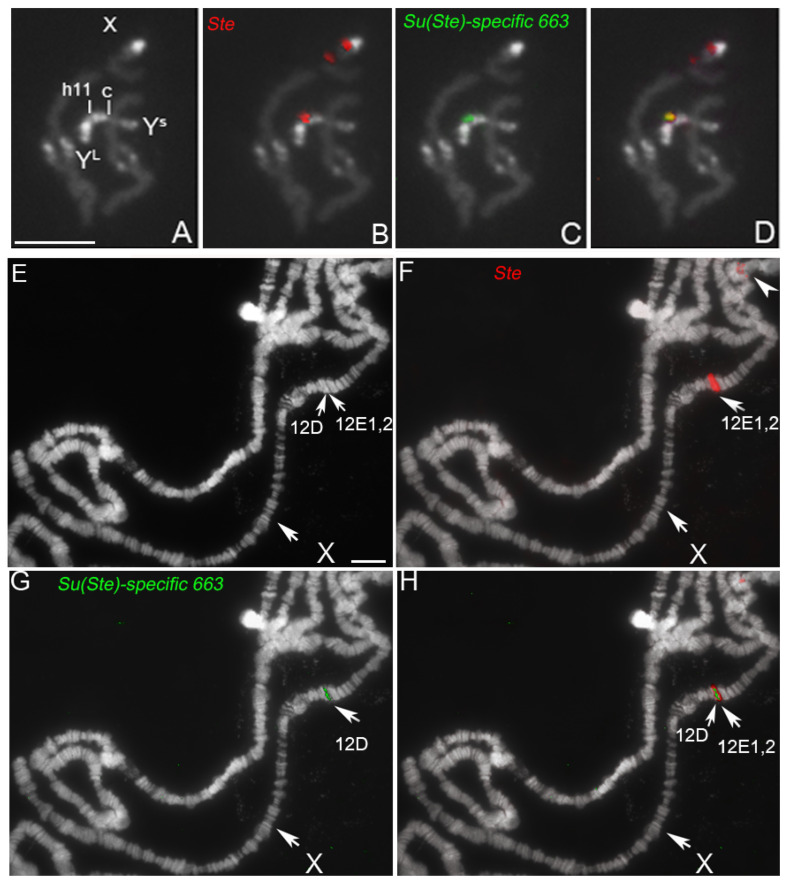
(**A**–**D**) Localization by fluorescence in situ hybridization of pUSTE6-Cy2 labeled (red) and *Su(Ste)*-specific 663-FluorX labeled (green), on mitotic chromosomes of *D. melanogaster*, stained with DAPI. (**B**) FISH signals of pUSTE6 on the chromosome X in the *Ste*^eu^ and *Ste*^het^ loci and in the h11 region on the Y chromosome; (**C**) FISH signal of *Su(Ste)*-specific 663 in the h11 region on the Y chromosome; (**D**) a merge of the two hybridization patterns; the yellow signal on the merged figure indicates the co-localization of the two probes. Scale bar indicates 5 µm. (**E**–**H**) Localization of the same probes on the polytene chromosomes; (**F**) hybridization signals of pUSTE6 in the chromocenter (arrowhead) corresponding to X heterochromatic *Ste* locus and (arrow) in the 12E1,2 euchromatic region; (**G**) *Su(Ste)*-specific 663 hybridization signal in the region between 12D and 12E1,2 bands; (**H**) a merge of the two hybridization patterns. Scale bar indicates 10 µm.

**Figure 4 cells-11-03725-f004:**
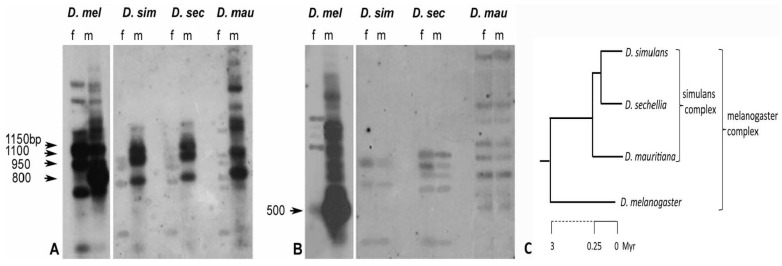
(**A**,**B**) Southern blotting analysis of the *Ste* (**A**) and *Su(Ste)*-specific 663 (**B**) distribution in males (m) and females (f) genomic DNA, digested with CfoI enzyme, in the Drosophila species indicated above. (**A**) In *D. melanogaster*, pUSTE6 probe highlights the expected three main 1150-, 1100-, and 950-bp fragments on the X chromosome and the 800-bp Y-linked fragment. In the closely related species analyzed, three main hybridization bands of different molecular weight are evident in males. (**B**) In *D. melanogaster* males, *Su(Ste)*-specific 663 probe hybridization highlights one main band of 500 bp repeated fragments and several additional bands, while only few, faint bands are highlighted in the simulans complex species. The filter was sequentially probed with ^32^P-labeled pUSTE6 and after label stripping, it was rehybridized with ^32^P-labeled *Su(Ste)*-specific 663 probe. (**C**) Phylogenetic relationships among the four species of the melanogaster complex.

**Figure 5 cells-11-03725-f005:**
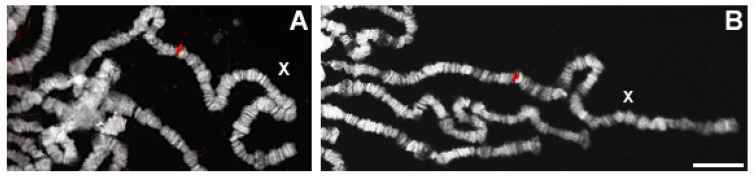
(**A**,**B**) Localization of crysi3 and crymau3 Cy3-labeled probes, on the polytene chromosomes of *D. simulans* (**A**) and *D. mauritiana* (**B**) stained with DAPI. Scale bar indicates 10 µm.

**Figure 6 cells-11-03725-f006:**
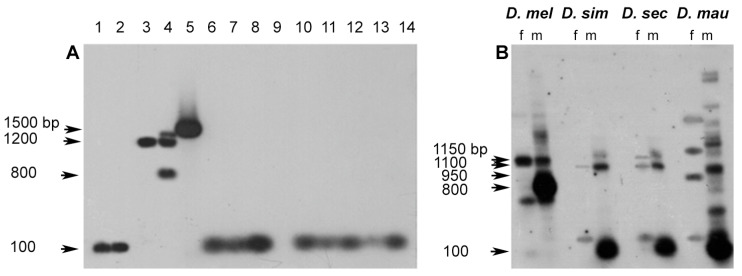
(**A**) Southern blotting analysis of lambda phage DNA, isolated from *D. simulans* library, digested with CfoI enzyme; hybridization of pUSTE6 probe identifies a fragment of 100 bp in most lambda phages, except in three. Phages loaded in lanes 2, 3, 4, and 5, named DsλF6, DsλF4, DsλF7, and DsλF20, respectively, were molecularly analyzed. (**B**) Southern blotting analysis of males (m) and females (f) genomic DNA of the indicated species, digested with CfoI enzyme, hybridized with a DsλF6 subclone (F6.3 clone).

**Figure 7 cells-11-03725-f007:**
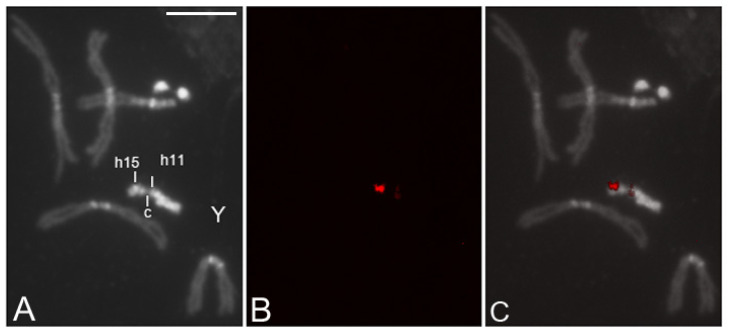
(**A**–**C**) Localization by fluorescence in situ hybridization of the F6.3 clone on the *D. simulans* mitotic chromosomes. Two signals map on the Y chromosome. The strong hybridization signal is located on the short arm of the Y chromosome, in the h15 band; a second faint signal maps in the h11 region, beside the centromere. Scale bar indicates 5 µm.

**Figure 8 cells-11-03725-f008:**
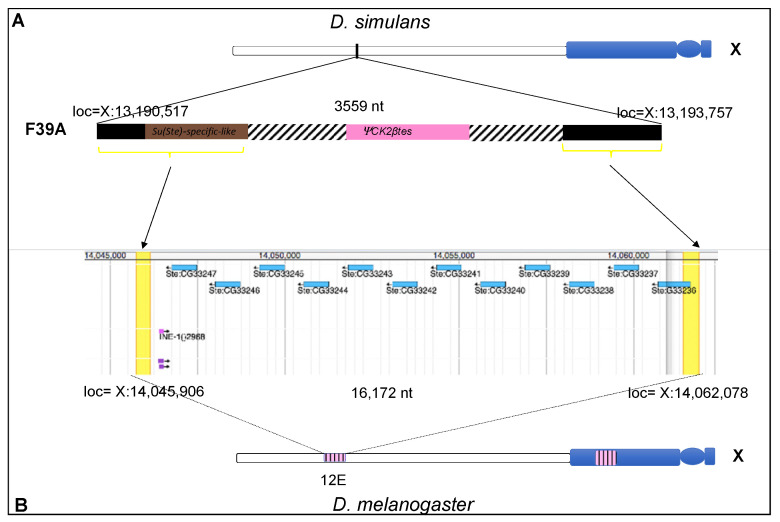
(**A,B**) Schematic of the F39A clone and its corresponding localization in the *D. simulans* and *D. melanogaster* X chromosome. In the X chromosome diagram, heterochromatin is depicted in blue, euchromatin in white, *Ste* and *Ste*-like in pink. (**A**) In the enlarged representation of the F39A clone are indicated, in brown, the *Su(Ste)*-specific-like region, and, in pink, the *Ste*-like corresponding to *ΨCK2βtes.* The nucleotide positions corresponding to both ends of F39A clone in *D. simulans* genome are indicated. (**B**) In the shaded part is represented the FlyBase Blast Report of JBrowse (*D. melanogaster*, r6.45) obtained from the alignment of the F39A sequence; the nucleotide positions are indicated. Black arrows indicate the localization of the orthologous sequences of both ends of F39A, with the syntenic X-chromosome region of *D. melanogaster*. The F39A clone corresponds to the region in which *Ste* locus is located in *D. melanogaster*. All drawings are not to scale.

**Figure 9 cells-11-03725-f009:**
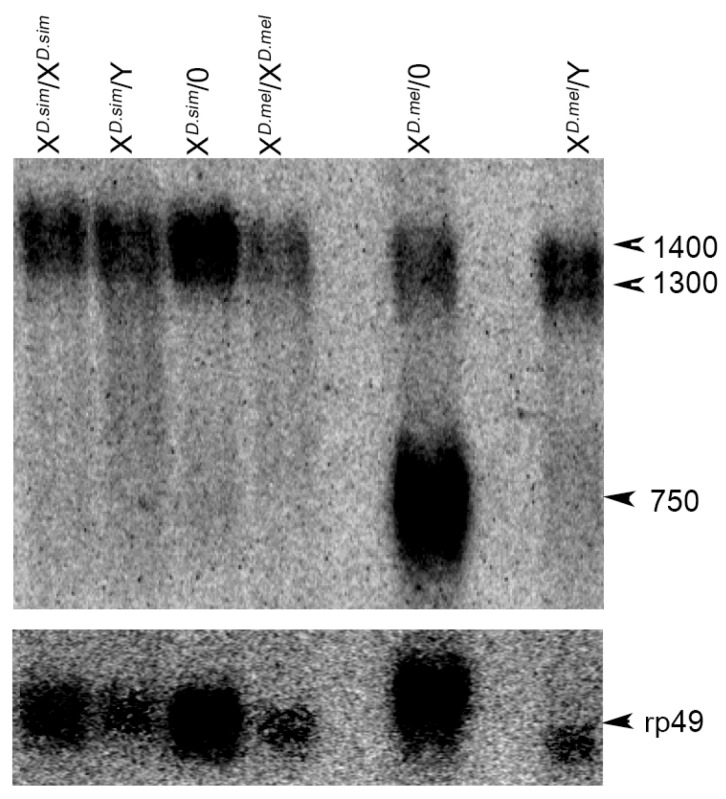
Northern blotting analysis of the *Ste*-like transcripts in *D. simulans*. Total mRNA extracted from adult ovaries and adult testes of the indicated genotypes probed with *Ste* cDNA antisense. No *Ste* transcript of 750 nt testes-specific is present in *D. simulans*; it is present in *D. melanogaster* Y*^cry−^* males only. The larger ubiquitous transcripts are present in females and in X/Y and X/0 males in both species. Molecular weight of transcripts is indicated; the transcript of ribosomal protein rp49 was used as a loading control.

## Data Availability

The data presented in this study are available on request from the corresponding author.

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
