# Peer review of "The *Drosophila simulans* Genome Lacks the *crystal*-*Stellate* System"

_cells, 2022, doi:10.3390/cells11233725_

Round 1

Reviewer 1 Report

The article is devoted to a very interesting topic - the evolution of the mysterious system of genetic interactions crystal-Stelate. In Drosophila melanogaster this system evolved as a set of homologous elements repeated many times in the genome. At the same time, the only vital function of such elements is the suppression of the expression of homologous sequences. That is, in general, the system works in the genome to suppress the expression of itself. In this case, the suppression is carried out by piRNA interference, which makes it a very convenient experimental object for studying the mechanism of this type of silencing and the evolution of the elements involved in it.

In the manuscript under review  in the introduction the authors provide an overview of what is known about the organization of loci associated with the crystal-stelate system in Drosophila melanogaster and the current ideas about the evolution of these loci. Despite the fact that the genomes of species related to D. melanogaster have been sequenced, and genome assemblies and some transcriptomic data are available, these data are not sufficient to conclude on the presence or absence of crystal-stellate systems in these species. This is due to the repeated nature of these elements and their rapid evolution.

In the presented work, the authors use the methods of classical molecular genetics - screening of phage genomic libraries, southern blot hybridization and northern blot hybridization in combination with the analysis of available genomic and transcriptome data. In my opinion, the authors have got convincing interesting results and  the article is worthy of publication in Cells. , but it needs some corrections. At the same time, some revision of the manuscript is needed.

Comments:

At the end of the introduction and in the discussion, the authors, in my opinion, do not sufficiently show the position of the data obtained in the article in the previously available picture of the evolution of the system under study. In the discussion, it is very important to describe in more detail the novelty of the results obtained. Now it is written as if the authors just once again confirmed the already known conclusions. The last chapter of the discussion is written completely without reference to the results obtained in the article. It would be appropriate to shorten this part, since it is not directly related to the results of the work.

In order to make the results of the Southern blot analysis more understandable, it would be very appropriate to provide schematic restriction maps of all Ste homologues in the melanogaster genome and all analyzed clones. The authors know all the DNA sequences and it should not be difficult.

The introduction is clear and concise, the diagram in Figure 1 is very informative, but I am missing another diagram illustrating the chapter “Evolution of the cry-Ste system” and including Y-linked CK2βtes and PCKR. Such a scheme would greatly help to understand the results obtained by the authors in the analysis of Ste homologues in other species.

In the title of the chapter “Overview of Ste- and Su(Ste)-like sequences in the melanogaster complex” and further in the text, the authors use the expressions: “melanogaster complex” and “Simulans complex” . If the authors want to combine D. melanogaster and D. simulans complexes, the more correct spelling is Drosophila melanogaster species subgroup, but such a subgroup includes other species. In any case, the simulance complex is not part of the melanogaster complex. It may be useful to add a phylogenetic tree (for example, to Figure 3) to make it more clear to the reader.

I cannot agree with the authors' interpretation of the mapping of Ste to the 12D3,4 polytene chromosome.

First, even when comparing Figure 2E and Figure 2G, it can be seen that the arrow labeled 12D3 matches the position of the arrow labeled 12E above.

Secondly, the bands 12D3-4 are drawn by Bridges between D1-2 and E1-2, these are so thin bands that they are not even visible on the electron microscopic map of Saura and Sorsa (https://wiki.flybase.org/wiki/flybase:maps). There is a very thin gap between D1-2 and E1-2, which cannot be resolved in the photo with the magnification presented in the manuscript. Perhaps a detailed framing would improve the picture, however I don't believe the signal will be in 12D for the following reason:

Third: The article by Belyaeva et al., 2012 presents the coordinates of regions of intercalary heterochromatin, established on the basis of genomic mapping of underreplication zones and analysis of the distribution of various chromatin proteins in the genome. For 12E, the predicted boundaries of the intercalary heterochromatin band were further confirmed by FISH. Comparison of these boundaries and the position on the genomic map of the Ste cluster show that the leftmost of the gene cluster Ste Ste:CG33247 ( chrX:14046772-14047495 Dm6 ) lies 50 kb from the distal border of the 12E1-2 band, and lies inside this band (chrX:13891496-14092986 Dm3 = chrX:13997463-14198953  Dm6)

And one more consideration in favor of the impossibility of localizing ste in 12D3-4. It has been shown that the Ste repeat attracts compactizer proteins and even causes the position effect on transgenes. It would be strange if such chromatin would correspond to a very open (by morphological criteria) chromatin in 12D3,4.

Even regardless of the correctness of the mapping it is necessary to unify the 12 D3, 12D3, 12D3,4

Load control for Northern blots looks worse than all other lines, some tracks are almost empty. What is the reason?  Has load control been done for Southern blots?

Line 20 “The organization and  expression of cry- and Ste-like sequences have been deepened in the D. simulans genome, “ not organization but knowledge on organization have been deepend

Lines 289-292 should be supplemented with the names of the primers used for PCR, otherwise the text looks contradictory: first, the authors receive a PCR product for three species, and then they write that the PCR reaction does not go. Probably, the authors mean reactions with different primers.

Line 353 please indicate which Drosophila species is  in question (whose Ssl gene)

Line 355 What is the “latest D. melanogaster genome”? Dm6?

Line 538  “on the Southern blotting analysis previously employed” Do the  authors mean - the same filter, not analysis?

For the results described in lines 355-357, it would be useful to give an alignment or alignment scheme in supplementary.

For lines 451-453 it would be interesting to know what is known about the origin of this transcript 1300-1400

Figures S2 and S3 are very small, they need to be enlarged.

I'm not a native speaker and can't claim to be an expert in English, but I've noticed sentences with the wrong word order, like: “Recently have been discovered on the Y chromosome, a source of testis-specific piRNAs, whose deletion  disrupt sperm maturation”. I would advise authors to contact a professional editor.

Reviewer 2 Report

The paper studied the crystal-stellate system in D. simulans. They found, unlike the D. melangoster, the crystal-stellate system is lacked in the D. simulans.

The design was appropriate, and the result was clear. The conclusion is interesting.

I only have one suggestion:

In the results part, the authors only describe the data without any analysis and did not give any conclusion base on the data. This made this part very difficult to read.

Author Response

The paper studied the crystal-stellate system in D. simulans. They found, unlike the D. melangoster, the crystal-stellate system is lacked in the D. simulans.

The design was appropriate, and the result was clear. The conclusion is interesting.

I only have one suggestion:

In the results part, the authors only describe the data without any analysis and did not give any conclusion base on the data. This made this part very difficult to read.

-Suggestions have been taken into account. In the Results, some statements were highlighted. Discussion has been amended and shortened.

Thank you so much for your valuable observation.

Reviewer 3 Report

This is a very descriptive story, showing that the repetitive heterochromatin-chromatins components Ste and Su(ste) are only found in D. melanogaster and not in sister species. This component makes offspring from D. melanogaster and D. simulans sterile, similar to offspring from D. melanogaster and D. mauritiana. The experimental design is solid, showing the absence of the full component in other species by fluorescence in situ hybridisation. However, the functional experiment is lacking, but some presented as “not shown” in the discussion. These images should be shown in the manuscript. Several conclusions are drawn from the discussion, making the experimental part preliminary.

Major concerns:

What is the evidence for the transcripts from the components act as male specific piRNA in D. melanogaster?

What is the natural function of the protein from the cry-STE, and what is different the cry from the Y-chromosome? Is it expressed at a high level and forms aggregates? What is the promoter like, testis –specific?

How does it interfere with male flies being fertile? Are crystalles formed?

Are not the pre-components transcribed or does it lack a promoter or is the heterochromatin silencing working in other species?

Experiments mentioned in the discussion, such as the changed morphology of testes of offspring of D. melanogaster and D. simulans.

I would advise the authors to use direct quotes in the text, since it makes these citations extremely important. Rephrase these sections. The discussion should also be shortened, it now contains discussion beyond the experimental data.

Round 2

Reviewer 1 Report

The authors took into account all my comments, so I believe that the article can be published in its current form.